# Enhancement of Phthalocyanine Mediated Photodynamic Therapy by Catechin on Lung Cancer Cells

**DOI:** 10.3390/molecules25214874

**Published:** 2020-10-22

**Authors:** Giftson J. Senapathy, Blassan P. George, Heidi Abrahamse

**Affiliations:** Laser Research Centre, Faculty of Health Sciences, University of Johannesburg, 2028 Doornfontein, South Africa; giftsons@uj.ac.za (G.J.S.); blassang@uj.ac.za (B.P.G.)

**Keywords:** catechin, photodynamic therapy, lung cancer, natural product, dose-response, phytochemical, cancer treatment

## Abstract

Worldwide, lung cancer remains one of the leading cancers with increasing mortality rates. Though chemotherapy for lung cancer is effective, it is always accompanied by unavoidable and grave side effects. Photodynamic therapy (PDT), using novel photosensitizers, is an advanced treatment method with relatively few side effects. Plant products are emerging as potent photosensitizers (PSs). The dose-dependent effect of Catechin (CA) (20–100 µM) on cellular morphological changes, cell viability, cytotoxicity, proliferation, DNA damage and apoptosis were studied on A549 adenocarcinoma alveolar basal epithelial cells. The effect of CA, along with Zinc phthalocyanine PS at 680 nm and 5 J/cm^2^ fluency was also studied. As the doses of CA increased, the results showed a pattern of increased cytotoxicity, accompanied by decreased cell viability and proliferation in A549 cells. Also, at 52 µM (IC_50_), CA in combination with PS significantly increased the cytotoxicity, DNA damage, and apoptosis, as compared to control and PS alone, treated cells in PDT experiments. These findings leave a possible thread that CA can be used in the application of phyto-photodynamic therapy of cancer in future.

## 1. Introduction

Cancer is a multi-factorial disease characterized by the uncontrolled growth of cells resulting in the formation of malignant tumors. Globally lung cancer leads all the other types of cancers in terms of its occurrence, severity, morbidity, and mortality rates [1]. It is estimated that lung cancer deaths may outnumber 3 million by 2035. Smoking, environmental pollution, and radiation are important causes of lung cancer. The usual treatment strategies, such as radiotherapy, surgery, and chemotherapy were not offering satisfactory results for lung cancer [2]. Systemic administration of chemotherapeutic drugs, minimal drug entry to the site of action, side effects of commonly used chemotherapeutic drugs to the extent of pausing the treatment till patient recovery were some of the notable disadvantages of lung cancer chemotherapy, making the practice less appealing. Non-surgical treatment strategies have evolved significantly in the recent decades and are employed in treating lung cancer. Therefore, a new alternative treatment that would address all or most of these issues would be an ideal cancer treatment method [3].

Photodynamic therapy (PDT), or phototherapy or non-ionizing radiation therapy introduced in 1970s, is one of the latest treatment strategies when compared to chemo and radiotherapy. Ever since its discovery, in the past few decades it was employed in the treatment of various malignant tumors including melanoma, breast, colon and lung cancers with less invasiveness [4]. Compared to other conventional treatments like radiotherapy, surgery, and chemotherapy, it is more selective and relatively less toxic with less or no damage to normal cells. PDT is comparatively safe and minimally invasive [5]. Three individual components, namely the light source of appropriate wavelength, oxygen (O_2_) and light-sensitive chemical agents known as photosensitizers are essential for PDT. PDT induces the activation of chemical photosensitizers by laser lights of specific wavelength to produce singlet oxygen (O_2_^•^) and reactive oxygen species (ROS) that can harm the cell membranes and components resulting in necrosis and/or apoptosis. But the clinically available photosensitizers were accompanied with few disadvantages like minimal photosensitivity, phototoxicity, and less selectivity of tumors, which leads to the search for new compounds which have photosensitizing effects or which can modulate PDT [6,7].

A vast array of natural compounds has shown anticancer activities either separately or in blend with other therapies. Many phytochemicals, particularly flavonoids, have recently reported to have photosensitizing properties and hence, can act as natural photosensitizers in PDT [8].

Catechin or (+) Catechin hydrate (CA) (Figure 1) is a non-toxic and less expensive flavonoid present predominantly in green tea and other dietary products, plants, berry fruits (strawberry, raspberry, gooseberry, and blueberry), grapes, pineapples, kiwi, cocoa and beverages such as red wine and beer [9]. A green tea preparation (1 g tea leaves) has 30–40% of CAs in its 0.2–0.4 g of dry materials [10]. The term CA is originated from ‘catechu’, a tannic juice obtained from boiling the extracts of *Mimosa catechu*. CAs are powerful antioxidants, but besides this, they also can act as pro-oxidants. Research evidence has shown that they exhibit selective genotoxic and cytotoxic activities, particularly in cancer cells, including breast and cervical cancer cells, but not in normal cells [11]. Since CAs utilize the multiple targets of cancer treatment, they are found to be extremely useful to treat multidrug-resistant tumors [12]. They also affect the molecular mechanisms of angiogenesis, regulatory signals of cell death and other targets of cancer [13,14,15,16]. CAs are reported to have direct interaction with molecular pathways by binding to phospholipids and proteins. As a result, they modulate the enzymes and regulate cancer signaling pathways [17].

Based on the above literature, the present study was designed to explore the dose-dependant anticancer effect of CA in A549 lung cancer cells. Also, a special focus was given to study the effect of CA in combination with the PS Zinc phthalocyanine in PDT.

## 2. Results

The morphological changes of treated and untreated A549 cells are displayed in Figure 2. Untreated control cells (a) showed no abnormal cellular morphological changes and maintained the shape of lung cancer cells. Treatment with different doses of CA (20, 40, 60, 80, 100 µM) demonstrated distinct morphological changes (round and detached non-viable cells floating in the culture media), inferring an increase in cell death with the increase in CA doses.

The results of trypan blue viability assay showed a dose-dependent reduction in cell viability significantly in lung cancer cells treated with increasing concentrations of CA (20, 40, 60 80, 100 µM) when compared to untreated control cells (Figure 3). A 50% inhibition (IC_50_) of cells was observed at 52 µM concentration of CA.

LDH leakage assay was used to assess the integrity of membranes determined through the estimation of LDH leaked in cell media due to the damaged membrane of cells, which in turn was measured at 490 nm. The cells which received CA at different doses (20–100 µM), significantly (*p* < 0.001) increased the cell membrane damage than the control cells (Figure 4). This reflected a dose-dependent increase in the cytotoxicity upon CA (20–100 µM) treatment. LDH levels measured are directly proportional to the cell’s damage.

The energy levels of the cells were measured by calculating the ATP levels of the cells. The energy levels in A549 cells were higher, as indicated by the elevated ATP levels in control cells (Figure 5). Cells that received different doses of CA (20–100 µM) established a significant decline in ATP luminescence as related to control and CA alone treated cells. The reduction in ATP luminescence indicates a reduction in cell proliferation by treatments.

Based on the above data obtained from the morphological analysis, cell viability, cytotoxicity, and cell proliferation, it is thus clear that CA expressed a linear dose-dependent cell growth inhibition with an increase in the doses (20–100 µM). The IC_50_ dose of CA (52 µM) was employed further in PDT experiments.

In the PDT experiments, the cell viability analysis by trypan blue showed that untreated control (group 1) cells did not show any loss of viability, which was slightly decreased in laser alone treated cells (group 2) (84.77%) (Table 1). The PDT group (group 3) showed a significant drop in viability (51.43%). Interestingly when CA (52 µM) was introduced in PDT, it significantly reduced the cell viability to 27.80%, indicating a pronounced cytotoxic effect due to the presence CA in group 4.

Similarly, in LDH leakage assay, the introduction of CA in PDT along with PS and laser presented a significant surge in cytotoxicity accompanied by cell membrane damage when referred to untreated, laser and PDT groups (Table 1).

The effect of CA on cell proliferation in PDT of A549 cells was displayed in Figure 6. The energy level of the untreated control cells was found to be higher when compared to treated groups. Laser treatment and PDT showed a substantial drop in cell proliferation and indicating the effectiveness of treatment. Whereas, when CA was introduced in PDT, a further significant reduction of cell proliferation was noted when compared to PDT and control groups.

Hoechst stain is a nuclear DNA specific blue stain permanently staining the DNA of the cells and hence it is used to evaluate the degree of DNA damage after the treatments. The Hoechst stain results (Figure 7) showed intact undisturbed dense oval nuclei indicating no or negligible DNA damage. Whereas on PDT and phyto-PDT treatments (combined treatment of CA with PS), the characteristic nuclear damage events such as nuclear shrinkage, nuclear retraction, and chromatic condensation were observed.

The CA-ZnPcSmix induced cell death was confirmed further by Annexin V-FITC/PI staining through flow cytometric analysis (Figure 8). In the untreated group (a), the viable cell count was 99.9%, with apoptotic and necrotic cell count relatively 0%. The percentages of live:apoptotic:dead cells in laser alone treated group (b) was in the ratio of 52:47:0.1. Whereas in PDT (c) and CA-PDT (d) groups, it was found to be 43:50:6.1 and 26:63:10, respectively. The apoptotic cells percentage was enhanced in PDT and Phyto-PDT groups with reference to control groups.

## 3. Discussion

PDT is presently viewed as a promising treatment alternative to conventional cancer therapies for various cancers. In PDT, three individual non-toxic elements such as light, O_2,_ and PS cause damage to the cells and tissues when they are combined. The better results from PDT are based on the generation of ROS, O_2_ yield, and molecular stability of the photosensitizing agent [18]. As the commercially synthesized PSs were accompanied with some degree of toxicities, plant-based PSs with low-level laser irradiations could offer a more promising area of cancer research. In recent years, phytochemicals are being increasingly recognized for potent complementary cancer treatments [19]. In the present study, the treatment of green tea CA in A549 cells showed a dose-dependent increase in the anticancer activity as assessed by the various biochemical assays. Also, it was observed that CA augmented the overall effect of PDT when treated together with the PS (Zinc phthalocyanine).

A decrease in cell viability in CA and PS treated cells observed in the present study, revealed the increased cytotoxicity due to the CA and PDT. CA induced cytotoxicity in cancer cells was previously reported in many experimental studies [20,21]. The oxidative stress in cancer cells was reported to cause membrane damage, which is a contributing factor for the cytotoxic effects observed in the present study. In PDT, the energy emitted from the excited PS was transferred to molecular O_2_ to ROS, which is cytotoxic to the cells leading to cell death. As reported by previous works in our lab, increased oxidative stress by plant extracts and phytochemicals reduced the intracellular ATP levels [22]. Yang et al. reported that light-activated CA caused increased DNA strand breakage resulting in the depletion of cellular NAD^+^ and ATP levels leading to cell death [23]. The nuclear damage observed in A549 cells by CA and PDT through Hoechst staining is in correlation with the observed morphological changes in the treated cells revealing cytotoxicity and cell death. Leo et al. reported the induction of nuclear damage by catechins in cancer cells [21]. The reduced ATP levels in CA and laser-treated cells (PDT groups) in the present study, is due to this reason.

Increased LDH levels leaked into the cell media, as observed in the present study in CA and ZnPCSmix treated cells, is due to the disruption of the cell membrane by the treatment. Green tea polyphenols are known to induce cancer cell membrane damage resulting in the release of LDH [24].

The phototoxic effects of CA and PDT, as indicated by assay results of LDH cytotoxicity, ATP proliferation assays, and trypan blue cell viability, led the cells to death by triggering apoptotic signaling pathways. CAs were reported to influence the cancer signal transduction pathways leading to cell death [24]. In vivo and in vitro studies have confirmed the inhibition of carcinogenesis by CAs at initiation, promotion, and progression stages. This complicated obstruction of tumorigenesis is due to the combined pharmacological effects of CA [25].

The -OH bond of CA molecule in the heterocyclic ring was excited and oxidized by photosensitization, resulting in the formation of a quinone derivative, which increases oxidative stress in cancer cells. CA could utilize these mechanisms in PDT, which would result in cancer cell damage [26]. Previously, Yang et al. reported the cytotoxic action of CA through aerobic photo-oxidation upon irradiation with blue light [27]. Numerous research evidence refers to the health-promoting capability of CAs with reference to its antioxidant activity. However, like other polyphenols, CAs can also generate as well as scavenge free radicals based on multiple factors. By a controlled combination of both mechanisms based on the requirements of the cell, CA may exert their beneficial effects, particularly in the fight against diseases like cancer [10,20,28].

PDT exerts its therapeutic effect mainly by the production of ROS resulting in increased oxidative stress in cancer cells. It is reported that, in PDT, the entry of PS to normal cells is either restricted or highly reduced due to the increased tumor vasculature and permeability of cancer cell membrane compared to normal cells. Moreover, ROS generated during PDT is short-lived and is effective within a very narrow range of 20 nm, thereby reducing its effect on normal cells. Thus, normal cells are not affected during PDT, which increases its clinical applications [8,29]. At the same time, CAs were also found to be protective over normal cells. The structure-activity relationship of CA was observed in previous experiments in relation to the phenomenon of photoactivation and photosensitization. The flavan-3-ol structure of CA shows the presence of two benzene rings with strong electron absorption upon UV light irradiation. In alkali or neutral solutions, CA was found to be sensitive to blue light resulting in photo-induced electron transfer. Upon irradiation, CA produces O_2_^•^ coupled with photosensitized oxidation, which could be harmful to the cellular components. The singlet molecular oxygen could react directly with substrates, which in turn react with oxygen to produce free radicals resulting in cellular damage [30].

Previous studies have reported that CAs improved the anticancer effects in vitro, in vivo, and in clinical trials when combined with other chemotherapeutic drugs. The photodynamic activity of other forms of green tea CAs has already been stated in several cancer types [31,32,33,34]. When combined with the photosensitizer Radachlorin at concentrations 2.5–5 µg, tea CAs showed increased anticancer activity in vitro and in vivo [31]. In a recent study by Niu et al. (2020), oral administration of tea CA to human volunteers during radiotherapy reduced the therapy-induced side-effects [35]. It was also reported that CAs synergistically combine with other chemical drugs by enhancing apoptosis in human lung cancer cells [36].

CA was previously reported to impede cell viability, proliferation, and induce apoptotic signals in a dose-responsive fashion in cancer cells [10]. The increased effect of CA on cell proliferation, membrane damage, nuclear DNA damage, and cytotoxicity in PDT, as evidenced in the present study, revealed its photoactivated cancer cell inhibitory effect. Hence, the results of the present study revealed the enhancement of overall Zn Phthalocyanine-PDT effects by CA, which confirms that this green tea flavonoid could have acted additively or synergistically when combined with a photosensitizer in the PDT of cancer. These findings instigate the possibility of introducing CA in phyto-photodynamic therapy of cancer in the future.

## 4. Materials and Methods

### 4.1. Chemicals and Reagents

(+) Catechin (CA) was obtained from Sigma, Johannesburg, South Africa (PHR1963). Fetal bovine serum (FBS; Biochrom, S0615), Roswell Park Memorial Institute 1640 medium (RPMI), (Sigma, R8758), penicillin/streptomycin (Sigma, P4333), amphotericin B (Sigma, A2942), trypan blue (Sigma Aldrich, T8154), phosphate-buffered saline (PBS), Hank’s balanced salt solution (HBSS) (Invitrogen, 10–543 F), propidium iodide (PI) and TrypLE Express (Gibco, 12563-029, Thermo Fisher Scientific, Johannesburg, South Africa), were used for the present study.

### 4.2. Cell Culture

A549 lung cancer cells used in this study were purchased from ATCC (ATCC^®^ CCL-185). A549 cells were maintained and cultured in RPMI medium complemented with 10% FBS, 0.5% penicillin/streptomycin, and 0.5% amphotericin B and incubated at 37 °C in 5% CO_2_ and 85% humidity. When the cells in the culture dishes were 80% confluent, the spent media were discarded, cells washed twice with HBSS and detached with TrypLE Express at a ratio of 1 mL/25 cm^2^. For the biochemical analysis, approximately 2 × 10^5^ cells were seeded in culture plates of diameter 3.4 cm^2^ with 3 mL complete medium and incubated for 4 h to attach.

### 4.3. Dose-Response Analysis of CA

A stock solution of 1 M CA was prepared in distilled water and stored at 4 °C. Further dilutions were made in culture media to obtain the desired concentrations as per the experimental design. For dose-dependant analysis, A549 cells were administered with increasing doses of CA (20, 40, 60 80, and 100 µM) for 24 h.

### 4.4. PDT Experiment

For PDT experiments, Zinc (II) phthalocyanine (ZnPCSmix), which is an II-generation PS, having a maximum light penetration in tissues at absorption Q bands between 520–770 nm was used in the present study. ZnPcSmix has a triplet quantum yield of 0.53, singlet oxygen quantum yield (FΔ) of 0.45, triplet lifetime of 2.95 s, and fluorescence quantum yield of 0.16 with an absorption peak at 680 nm [22]. Briefly, the cells were separated into 4 groups (Table 2); group 1 control cells received no treatment and irradiation. Group 2 cells received laser irradiation only, served as laser control. Cells in group 3, were treated with PS (15 µM), and group 4 cells were treated with PS (15 µM) and CA (52 µM-IC50 concentration). After 20 h of incubation (37 °C with 5% CO_2_), the cells in groups 2–4 were irradiated in the dark with the lids off using a diode laser of emitting wavelength 680 nm (National Laser Centre, Pretoria, South Africa) at 5 J/cm^2^ fluency [22]. The Laser parameters are summarized in Table 3. To avoid light interference from disturbing the PDT experiments, all the irradiations were done in the dark. The cells were further incubated in the dark for 24 h after irradiation and used for various biochemical assays.

### 4.5. Morphological Cellular Analysis by Inverted Light Microscope

The morphological changes in the cells were examined after 24 h post-treatment and recorded in the control and experimental groups using a camera attached inverted light microscope (CKX41, Olympus, Wirsam, Richmond, Johannesburg, South Africa) connected to the cellSens image analysis software. The harvested cells were resuspended in PBS or HBSS for further biochemical analysis.

### 4.6. Cellular Viability Analysis by Trypan Blue Assay

The cell viability was estimated using the trypan blue (Sigma-Aldrich T8154) dye exclusion test. The dye was taken in by the dead cells (appear blue) due to membrane damage by treatment drugs whereas the intact membrane in the live cells (appear white) will not allow the dye inside the cell. The live and viable cells selectively exclude the dye and this principle was employed for measuring cell viability. Equal amount (10 µL) of cell suspension and dye was mixed and added (in an aliquot of 10 µL) to a countess slide designed for Countess automated cell counter (Countess^®^ Automated Cell Counter, ThermoFisher Scientific, Johannesburg, South Africa). The number of live, dead and total cells present was counted automatically by inserting the slide into the countess cell counter and the percentage cell viability was calculated.

### 4.7. Cell Proliferation by ATP Luminescence Assay

The measure of energy carrier molecule ATP released from the active cells was calculated by CellTiter-Glo^®^ ATP luminescence assay kit (Promega, G7570, Anatech Analytical Technology, Bellville, South Africa). The cell proliferation is calculated based on the amount of metabolically active cells retained after treatment. Equal volumes of (50 µL) of ATP CellTiter-Glo^®^ reagent cell suspension was added to an opaque-walled 96 well plate (BD Biosciences, 353,296, San Jose, CA, USA) and mixed in an orbital shaker for 10 min in the dark at room temperature (RT). The luminesce released was measured with media as background control through a Multilabel Counter (Perkin Elmer, VICTOR3™, 1420, Separation Scientific, Johannesburg, South Africa) in Relative Light Units (RLU).

### 4.8. Membrane Integrity Assessment by Lactate Dehydrogenase (LDH) Assay

The CytoTox96^®^ Assay kit (Promega S.A. G4000, Whitehead Scientific, Johannesburg, South Africa) was used to measure the LDH released by treated and untreated cells. The enzyme LDH is present in the cytosol of live cells that is released in the culture media due to membrane damage forming a red formazan complex by reacting with the assay fluid, which was read at 490 nm. Hence, the measure of LDH diffused in media gives the measure of membrane damage or cytotoxicity. Briefly, equal volumes of (50 µL) substrate and spent media from each group were added, mixed, and incubated for 30 min in the dark at RT for the formation of formazan complex, which was read using a Multilabel Counter.

### 4.9. Assessment of Nuclear Damage by Hoechst Stain

After 24 h of drug and PDT treatments to the cells grown in coverslip, they were washed twice with PBS (pH 7.4), fixed with 4% paraformaldehyde, and stained with 10 mg/mL Hoechst stain (Hoechst 33258, H21491, Invitrogen, ThermoFisher Scientific, South Africa) for 15 min. The stained cells were washed with PBS (pH 7.4) and viewed for nuclear damage was viewed in Olympus BX41 fluorescent microscope, South Africa.

### 4.10. Analysis of Cell Death by Annexin V-PI Stain

The percentage of non-apoptotic and apoptotic cells in the control and treated groups through flow cytometric analysis using Annexin V- fluorescein isothiocyanate (FITC) kit (Becton Dickinson, 556570, Scientific Group, Johannesburg, South Africa). Briefly, an aliquot of 100 µL (1 × 10^6^ cells/mL) of cell suspension was washed twice with 1xPBS (pH 7.4), suspended in 1x binding buffer and stained with Annexin V-FITC (5 µL) and PI (5 µL) for 15 min (dark) at RT. The population of live, apoptosis, and necrotic cells were determined using Fluorescence-Activated Cell Sorting (FACS) Aria flow cytometer (Becton Dickinson).

### 4.11. Statistical Analysis

Lung cancer cells from passages 16–20 were employed for the present study. All the experiments were carried out at least 4 times (*n* = 4) and the results were statistically analyzed by SigmaPlot version 13 using the student’s paired t-test. Values are represented as mean± standard error (SE) at a statistical significance of *p* < 0.05 (*), *p* < 0.01 (**) or *p* < 0.001 (***).

## 5. Conclusions

Based on the previous literature, we found that the effect CA in PDT of A549 lung cancer cells was not reported. The overall results of the study revealed that CA promoted cell death in lung cancer cells in a dose-dependent manner, and it also offered better results in combination with PS in PDT. The individual dose of the anticancer drug is one of the very important factors determining the efficiency of cancer therapy. Reports suggest that CA synergistically combines with other anticancer compounds and treatments. This leaves a chance for combining CA with ZnPcSmix as a novel treatment strategy to improve PDT outcomes. This combination could also possibly reduce the optimum treatment doses of the commercial photosensitizers, which in turn could reduce the dose-limiting side effects of PDT. It should be important to note that since the tissue concentrations of these flavan-3-ols are in the micromolar range, a targeted nano-drug delivery system with CA would offer many promising results in cancer PDT in the near future.

## Figures and Tables

**Figure 1 molecules-25-04874-f001:**
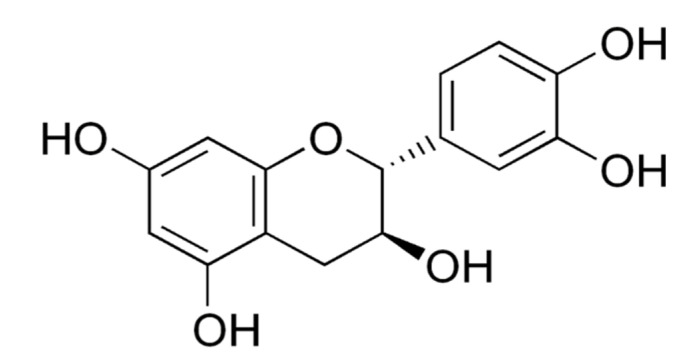
Chemical structure of catechin or (+) catechin hydrate.

**Figure 2 molecules-25-04874-f002:**
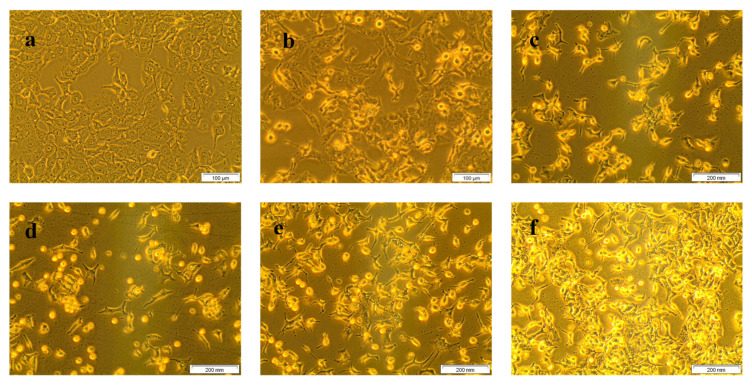
Morphological changes of A549 cells due to the result of CA treatment in different doses (40× magnification). Where, (**a**) are untreated cells; (**b**–**f**) are cells were treated with 20, 40, 60, 80, and 100 µM of CA, respectively. Morphological changes, including rounding of cells, detachment, floating cells in culture media, and brightly illuminated dead cells were observed in the images. It was clear that a steady increase in the aforesaid morphological changes identifying cancer cell death were noted in CA treated cells (**b**–**f**) with reference to untreated control cells revealing the dose-dependent cell growth inhibition of CA.

**Figure 3 molecules-25-04874-f003:**
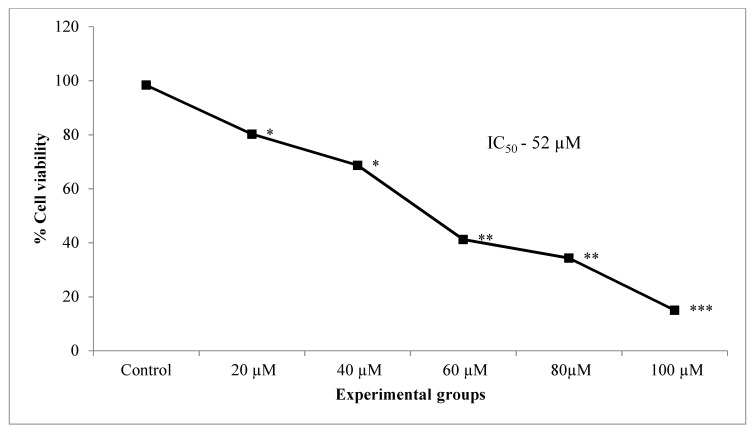
Effect of different doses of CA (20–100 µM) on the percentage cell viability of A549 lung cancer cells. Values presented as mean ± SE from independent duplicate experiments; *n* = 4. * *p* < 0.05, ** *p* < 0.01 and *** *p* < 0.001.

**Figure 4 molecules-25-04874-f004:**
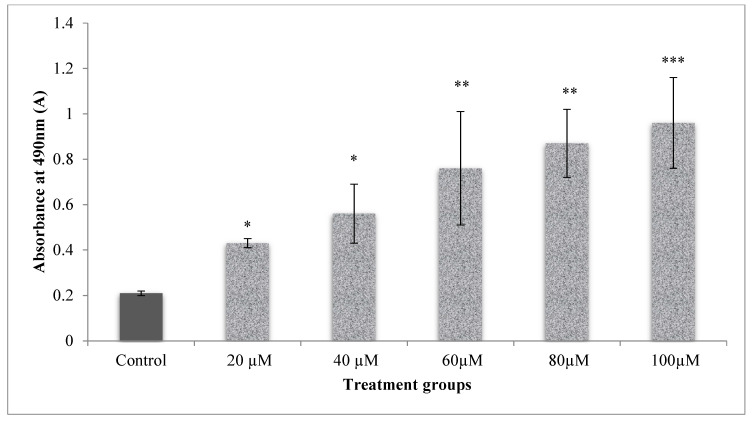
Dose-dependent cytotoxic effect of CA on A549 lung cancer cells as measured by LDH leakage. The test samples showed increased cytotoxicity with an increase in doses when control compared to control samples. Significant dose-dependent cytotoxicity was observed in A549 cells treated with CA (20–100 µM). Values presented as mean ± SE from independent duplicate experiments; *n* = 4. * *p* < 0.05, ** *p* < 0.01 and *** *p* < 0.001.

**Figure 5 molecules-25-04874-f005:**
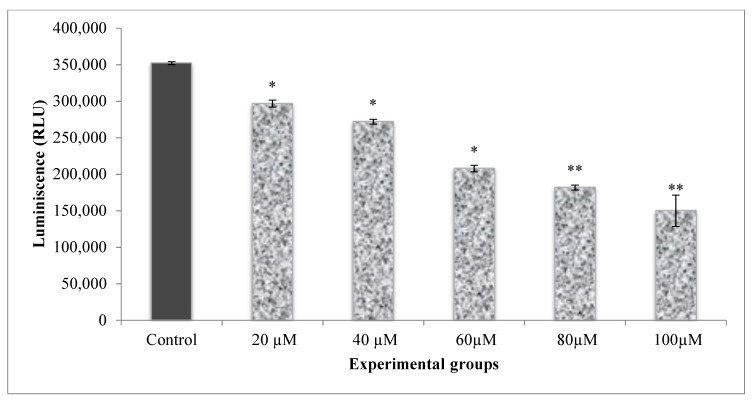
Dose-response analysis of CA on A549 cell proliferation as measured by cellular ATP. The amount of cellular ATP was estimated in terms of RLUs. A dose-dependant decrease in the ATP luminescence corresponding to the cell proliferation in CA treated A549 cells was observed. Values presented as mean ± SE from independent duplicate experiments; *n* = 4. * *p* < 0.05, ** *p* < 0.01.

**Figure 6 molecules-25-04874-f006:**
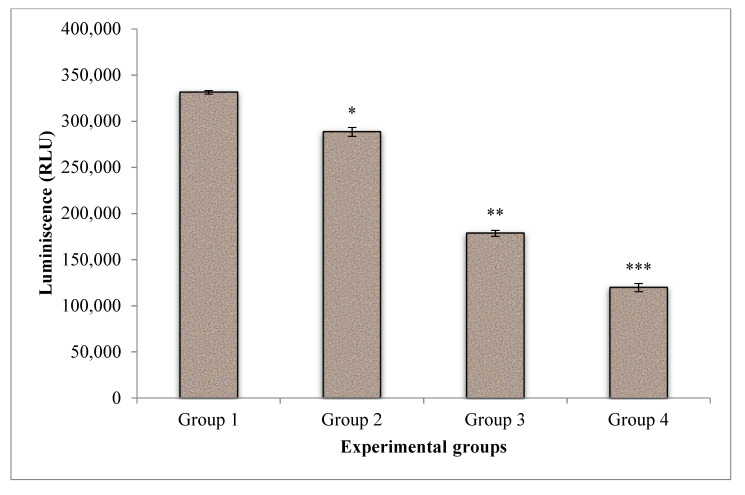
Effect of CA on cell proliferation of A549 cells in PDT of lung cancer cells. The amount of cellular ATP was estimated in terms of Relative Light Units (RLU). A dose-dependant decrease in the ATP luminescence corresponding to the cell proliferation in CA treated A549 cells was observed. Group 1 (Control), untreated cells; group 2, laser control (5 J/cm^2^); group 3 (PDT), PS (15 µM) + laser (5 J/cm^2^); group 4 (Phyto-PDT), CA (52 µM) + PS (15 µM) + laser (5 J/cm^2^); Values presented as mean ± SE from independent duplicate experiments; *n* = 4. * *p* < 0.05, ** *p* < 0.01 and *** *p* < 0.001.

**Figure 7 molecules-25-04874-f007:**
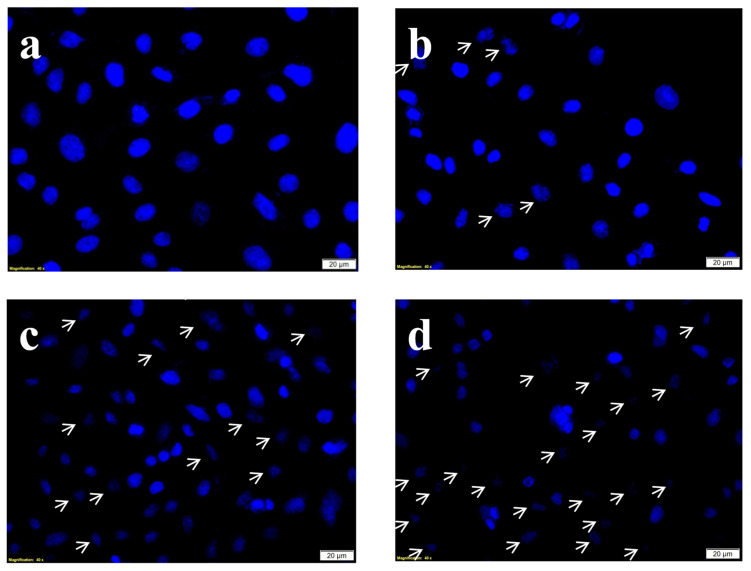
Effects of CA and PDT on nuclear DNA damage of A549 cells by DNA-specific Hoechst stain. Where, (**a**) group 1 (Control), untreated cells; (**b**) group 2, laser control (5 J/cm^2^); (**c**) group 3 (PDT), PS (15 µM) + laser (5 J/cm^2^); (**d**) group 4 (Phyto-PDT), CA (52 µM) + PS (15 µM) + laser (5 J/cm^2^) The arrow marks indicate the nuclear changes including cellular retraction, nuclear shrinkage and nuclear condensation which are characteristic signs of nuclear damage. The nuclear damage was increased in CA + PDT treatment when compared to PDT, laser alone, and control groups.

**Figure 8 molecules-25-04874-f008:**
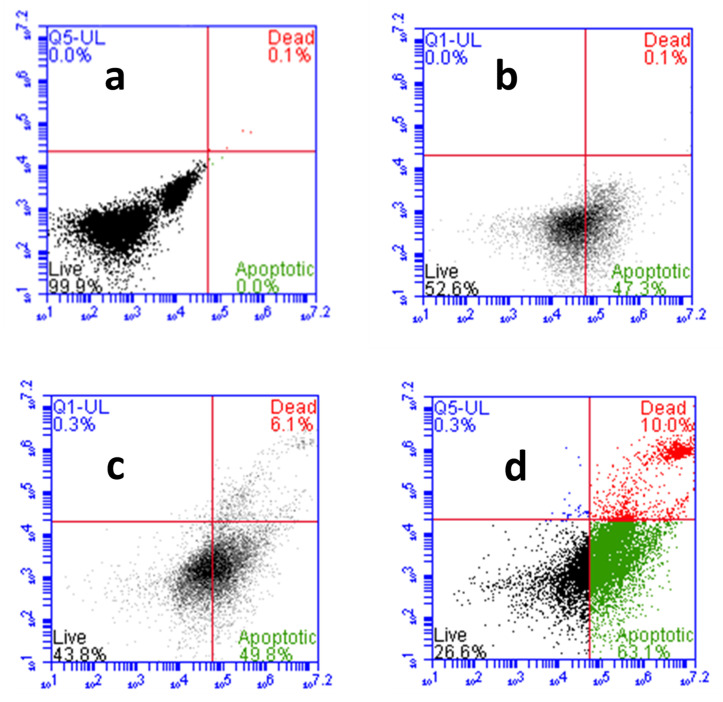
Effects of CA and PDT on cell death of A549 cells by Annexin V-FITC/PI staining. Where, (**a**) group 1 (control), untreated cells; (**b**) group 2, laser control (5 J/cm^2^); (**c**) group 3 (PDT), PS (15 µM) + laser (5 J/cm^2^); (**d**) group 4 (Phyto-PDT), CA (52 µM) + PS (15 µM) + laser (5 J/cm^2^). CA-PDT treatment resulted in significantly increased number apoptotic and necrotic cells compared to PDT, irradiation alone and control cells.

**Table 1 molecules-25-04874-t001:** Effect of CA in PDT on percentage cell viability and cytotoxicity of lung cancer cells.

Groups	Cell Viability (Trypan Blue)	Cytotoxicity (LDH Leakage)
	Percentage (%)	Absorbance @490 nm
Group 1	98.24 ± 0.84	0.08 ± 0.02
Group 2	84.77 ± 0.39 *	0.34 ± 0.01 *
Group 3	51.43 ± 0.72 **	0.76 ± 0.14 **
Group 4	27.80 ± 1.56 ***	1.53 ± 0.52 ***

Group 1 (Control), untreated cells; group 2, laser control (5 J/cm^2^); group 3 (PDT), PS (15 µM) + laser (5 J/cm^2^); group 4 (Phyto-PDT), CA (52 µM) + PS (15 µM) + laser (5 J/cm^2^); Values presented as mean ± SE from independent duplicate experiments with statistical significance at; *n* = 4; * *p* < 0.05, ** *p* < 0.01 and *** *p* < 0.001.

**Table 2 molecules-25-04874-t002:** Experimental design for the effect of CA in PDT of A549 cells.

Groups	Treatment Schedules
Group (Control)	Untreated A549 cells (2 × 10^5^ cells); 0 J/cm^2^ Fluence
Group 2 (Laser control)	Cells received irradiation only (5 J/cm^2^)
Group 3 (PDT)	Cells received PS (15 µM) and irradiation (5 J/cm^2^)
Group 4 (Phyto-PDT)	Cells received CA (52 µM), PS (15 µM) and irradiation (5 J/cm^2^)

**Table 3 molecules-25-04874-t003:** Laser parameters employed in the present study.

Parameters	
Laser type	Semiconductor (diode)
Wavelength	680 nm
Wave emission	Continuous
Fluence	5 J/cm^2^
Output power	200 mW/cm^2^
Irradiation time	3 min, 26 s
Spot size	9.1 cm^2^

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
