# Peer review of "Enhancement of Phthalocyanine Mediated Photodynamic Therapy by Catechin on Lung Cancer Cells"

_molecules, 2020, doi:10.3390/molecules25214874_

Round 1

Reviewer 1 Report

Manuscript entitled “Enhancement of Phthalocyanine mediated photodynamic therapy by Catechin on lung cancer cells” provides the scientific novelty of the research, however, before publication, several comments must be implemented:

Introduction: lines 71-72, authors describe specific mechanisms of anti-cancer modes of action, therefore, it is very appropriate to cite specific articles, used reference n.12 is too general! I suggest to authors to consider these recent references to cite the anti-agiogenic and apoptotic activities of flavonoids, including catechin:

  • J Cancer Res Clin Oncol. 2020 Sep 9.
  • Cancers (Basel). 2018 Dec 28;11(1):28.

Results:

Fig.2, please insert letters a-f into images and thus specify individual treatments,

Fig.4 does not show absorbance in 100 uM,

Line 133, please check value 89.03%, because Table 1 shows value 82.03%,

Line 133, correct the numbering - Table 1,

Authors numbered three Figures as Fig. 6, (shown in pages 6, 7, 8). Moreover, quote Figs. 6-8 into the text.

The additive increasing of nuclear damage mentioned in lines 165-66 (groups: laser alone, PS+laser, CA+PS+laser) is not seen in Fig.7, page 7. In this regard, please use more representative images b, c, and d.

Discussion: Shortly mention/discuss, how PDT can affect the oxidative stress in normal cells. Such discussion is very important for clinical cancer research.

Author Response

Reviewer 1 comments

Author Response

Manuscript entitled “Enhancement of Phthalocyanine mediated photodynamic therapy by Catechin on lung cancer cells” provides the scientific novelty of the research, however, before publication, several comments must be implemented:

Introduction: lines 71-72, authors describe specific mechanisms of anti-cancer modes of action, therefore, it is very appropriate to cite specific articles, used reference n.12 is too general! I suggest to authors to consider these recent references to cite the anti-agiogenic and apoptotic activities of flavonoids, including catechin:

·         J Cancer Res Clin Oncol. 2020 Sep 9.

·         Cancers (Basel). 2018 Dec 28;11(1):28.

Results:

Fig.2, please insert letters a-f into images and thus specify individual treatments

Fig.4 does not show absorbance in 100 uM,

Line 133, please check value 89.03%, because Table 1 shows value 82.03%

Line 133, correct the numbering - Table 1,

Authors numbered three Figures as Fig. 6, (shown in pages 6, 7, 8). Moreover, quote Figs. 6-8 into the text.

The additive increasing of nuclear damage mentioned in lines 165-66 (groups: laser alone, PS+laser, CA+PS+laser) is not seen in Fig.7, page 7. In this regard, please use more representative images b, c, and d.

Discussion: Shortly mention/discuss, how PDT can affect the oxidative stress in normal cells. Such discussion is very important for clinical cancer research.

Based on reviewer’s suggestion new references 12-15 were added in lines 71-72 and the numbering of references were corrected throughout the manuscript.

The reference- Cancers (Basel). 2018 Dec 28;11(1):28, included as suggested.

Letters ‘a-f’ – were inserted in figure 2.

In figure 4, the absorbance for 100 uM, was included and the graph was corrected.

Authors repeated the viability assay and inserted the new viability percentage values (line 133-136).

Table number corrected

In pages 6-8, the figure nos 6, 7 and 8 were corrected and the same was quoted in the text (line number 156, 172 and 183).

The representative images (Fig 7) of groups b, c & d were changed (line number 161-162).

A short discussion on the effect of PDT on normal cells was included at page 9 on lines 231-236.

Reviewer 2 Report

I write you in regards to manuscript entitled “Enhancement of Phthalocyanine mediated photodynamic therapy by Catechin on lung cancer cells” which you submitted to Molecules.

As author notes in this report, this study might be useful information for the application of phyto-photodynamic therapy of cancer. However, several revisions of manuscript are needed before it can be accepted for publication.

Major comments

・In combination study, CA was added at IC50 concentration. The cell viability in the group 3 and 4 are 46% and 31%, respectively (Table 1). Considering these results, I think that CA has not sufficiently enhanced the therapeutic effect of PDT. Is it unlikely that CA scavenge free radicals generated by PDT?

Minor comments

・In figure 2, the sub-numberings of figures were forgotten (a-f).

・In figure 6, I think that Hoechst image of the CA alone group should be showed. Considering cell viability (31%), the nucleus change was very little in the group 4. What do you think about this?

・Page 12 Line 336: Statistical test should be performed by nonparametric multiple comparisons.

・Page 12 Line 339: You mentioned that values are represented as mean ± standard deviation (SD). However, you mentioned that values are represented as mean ± error (SE) deviation figure legends. Which is correct?

Author Response

I write you in regards to manuscript entitled “Enhancement of Phthalocyanine mediated photodynamic therapy by Catechin on lung cancer cells” which you submitted to Molecules.

As author notes in this report, this study might be useful information for the application of phyto-photodynamic therapy of cancer. However, several revisions of manuscript are needed before it can be accepted for publication.

 Major comments

In combination study, CA was added at  IC50 concentration. The cell viability in the group 3 and 4 are 46% and 31%, respectively (Table 1). Considering these results, I think that CA has not sufficiently enhanced the therapeutic effect of PDT. Is it unlikely that CA scavenge free radicals generated by PDT?

 Minor comments

In figure 2, the sub-numberings of figures were forgotten (a-f).

In figure 6, I think that Hoechst image of the CA alone group should be showed. Considering cell viability (31%), the nucleus change was very little in the group 4. What do you think about this?

Page 12 Line 336: Statistical test should be performed by nonparametric multiple comparisons.

Page 12 Line 339: You mentioned that values are represented as mean ± standard deviation (SD). However, you mentioned that values are represented as mean ± error (SE) deviation figure legends. Which is correct?

The difference in cell viability between groups 3 and 4 was 15% and significant. However, based on reviewer’s suggestion, the cell viability assay was repeated, and the updated viability percentage was added (page 5; line 141-142; Table 1).

The sub-numberings (a-f) were inserted in figure 2.

In figure 6, the Hoechst images for groups b, c and d were changed as per the suggestion.

The standard statistical analysis of previously published works in our lab was adopted in this study. However, authors will consider this suggestion in our future research.

In page 12, line 349 ‘standard deviation’ was corrected as ‘standard error’

Round 2

Reviewer 2 Report

The manuscript has been revised well. I think this manuscript will be acceptable after style check.